# The Role of Previous History of Muscle Wasting in Burn Outcomes—A Burn Care Quality Platform Study

**DOI:** 10.3390/ebj6040061

**Published:** 2025-12-03

**Authors:** Elizabeth Blears, Jagger Godarzi, Sharon Shania, Krish Kondisetti, Julie Caffrey, Andrew J. Murton

**Affiliations:** 1Department of Plastic and Reconstructive Surgery, Tower Health–Reading Hospital, 420 S 5th Ave, West Reading, PA 19611, USA; 2Drexel University College of Medicine, 3141 Chestnut St, Philadelphia, PA 19104, USA; jg3938@drexel.edu (J.G.); ss5586@drexel.edu (S.S.); 3Philadelphia College of Osteopathic Medicine, 4170 City Ave, Philadelphia, PA 19131, USA; sk3483@pcom.edu; 4Bayview Medical Center, Johns Hopkins University, 4940 Eastern Ave, Baltimore, MD 21224, USA; jcaffre5@jhmi.edu; 5Sealy Center on Aging and Department of Surgery, University of Texas–Medical Branch, 301 University Blvd, Galveston, TX 77555, USA; ajmurton@utmb.edu

**Keywords:** burns, sarcopenia, cachexia, regression, malnutrition

## Abstract

Background: Burn patients can suffer prolonged hospital stays, infections, and wound breakdown. Given the complexity of burns, it is often difficult to determine which underlying factors contribute to complications. The Burn Care Quality Platform (BCQP) is the largest database of burn patients globally available, and it accounts for underlying or coinciding disease conditions present in burn patients. Muscle wasting conditions, such as sarcopenia, cachexia, and protein malnutrition, are suspected of causing worse outcomes. Prior analysis of BCQP data (2000–2017) demonstrated that patients with muscle wasting had prolonged hospitalization and adverse outcomes. Methods: Building on our previous work, we extended logistic regression analysis to BCPQ data through 2022 to assess whether reporting and outcomes had changed. Results: Updated BCQP data demonstrated a statistically significant increase in mortality in cachexia vs. non-muscle wasting patients (Odds Ratio [OR]: 2.2 [95% Confidence Interval (CI): 1.3–3.7], *p* = 0.004), but no increase in mortality was seen with protein malnutrition (OR: 1.1 [95% CI: 0.93–1.35], *p* = 0.239). However, the diagnosis rate of muscle wasting conditions decreased by 53% since the previous analysis, suggesting a potential under-reporting of these diagnoses in BCQP patients. Conclusions: Burn care could be augmented by better diagnosis of underlying conditions that predispose to muscle wasting.

## 1. Introduction

Burn injuries are among the most severe types of traumas, causing around 180,000 deaths annually worldwide from fires alone [1,2,3,4]. One of the greatest sources of morbidity in patients with severe burns, such as those covering > 30% of the total body surface area (TBSA), is hypermetabolism [5,6,7]. Hypermetabolism after burns is characterized by an elevated metabolic rate that can significantly deplete muscle protein [5,8]. In this state of hypermetabolism, patients often experience a profound proteolysis of muscle contractile proteins to generate gluconeogenic substrates to support the increased energy demands experienced by these individuals [9]. This skeletal muscle loss has profound long-term consequences, such as impaired mobility, chronic immunosuppression, and endocrine dysregulation [5,6,7,8,9].

Compounding the complex changes in metabolism from the burn itself are other types of preexisting muscle wasting, which are likely overlooked as a source of morbidity in the burn population [1]. Protein malnutrition, one cause of muscle wasting, is defined as a deficiency in dietary protein intake that leads to inadequate synthesis of muscle as well as weakened immunity and healing capacity [10]. Moreover, cachexia, the depletion of muscle with or without reduction in fat mass, is caused by underlying chronic illnesses, such as cancer or autoimmune disease [10]. Finally, sarcopenia, an additional muscle wasting condition, is commonly defined as the loss of muscle that results from the aging process rather than lack of protein calorie intake or an underlying illness [1,11]. In this analysis, each condition was defined strictly by its ICD coding criteria (Table A1): protein malnutrition = E43–E46; cachexia = R64; sarcopenia = M62.50. No laboratory thresholds were applied because such parameters are not available in the BCQP. While clinical overlap is recognized, the use of ICD-based definitions provides a standardized and reproducible approach for distinguishing these conditions in large-scale datasets. What impact these distinct causes of muscle wasting have on the treatment trajectory of burn patients is unclear.

Previously, the Burn Care Quality Platform (BCQP), the largest available global dataset of burn patients, was analyzed using data available from 2000 to 2017 to try to define the role of muscle wasting conditions in burn outcomes [1]. In this analysis, muscle wasting was defined as all available diagnoses within the International Classification of Disease (ICD) codes linked to muscle wasting, including protein malnutrition, cachexia, and sarcopenia. These conditions were found to be significant risk factors for a number of adverse clinical outcomes, including increased number of surgical procedures, longer hospital stays, and need for discharge to an assisted living facility rather than to home independent living [1].

However, in this earlier database analysis, muscle wasting was not found to be clinically significant in predicting mortality after adjusting for other confounding conditions, such as age, inhalation injury, and size of burn [1]. Additionally, only 2.6% of the patients were diagnosed with any type of preexisting muscle wasting condition, such as protein malnutrition, cachexia, and sarcopenia. Given the evolving awareness of these conditions and the fact that these rates were below baseline population rates, it was suspected that these conditions may have been under-reported, despite their clinical significance. With increased availability of data from the updated BCQP, which enrolled additional patients from 2017 to 2022, we sought to investigate whether or not attention to muscle wasting conditions had changed amongst burn clinicians and whether or not these conditions continued to play an important role in key clinical outcomes.

## 2. Materials and Methods

The analysis was performed after approval from the Johns Hopkins University Institutional Review Board under protocol #00416807. The BCQP was used to stratify patients into groups with “Sarcopenia,” “Cachexia,” and “Protein Malnutrition” using ICD codes (see Table A1). A patient group combining these diagnoses, called the “muscle wasting” group, was also created, similar to prior analysis [1]. Patients who had more than one muscle wasting condition were only counted once in the “muscle wasting” group. In the BCQP, ICD codes do not have an associated timing of diagnosis, making it difficult for the user to distinguish whether or not a given muscle wasting condition is preexisting or acquired during the acute stay of his or her burn care. The most recent BCQP was used to analyze the impact of these conditions on burn outcomes, which included patients from 2000–2022. Although ICD-10-CM codes were nationally implemented in 2015, the specific code for sarcopenia (M62.84) was introduced in 2016, and documentation of sarcopenia and cachexia (R64) appears in later BCQP records. Because earlier years showed minimal or inconsistent coding for these diagnoses, we limited muscle wasting analyses to 2017–2022 while retaining the full 2000–2022 BCQP cohort for context. Some patients with undiagnosed or miscoded muscle wasting conditions before 2017 were likely categorized in the non-muscle wasting group, reflecting under-recognition of these diagnoses during earlier registry years.

Similar to the previous analysis, the primary outcome of this study was mortality, and secondary outcomes included likelihood of being discharged home to independent living (rather than discharged to skilled nursing facility, long-term care facility, home with supportive nursing care, or death), development of sepsis, wound infection and/or skin graft failure during the acute burn treatment phase, as well as length of inpatient stay, length of intensive care unit (ICU) stay, length of ventilator support, and total number of skin debridement procedures required during acute burn care. Sepsis was defined using ICD codes as specified in Table A2. To enhance transparency and allow reproducibility of our methodology, extended variable definitions, coding criteria, and supplementary analyses are provided in the Appendix A. Only material essential for interpretation of the primary and secondary outcomes has been retained within the manuscript, while additional non-essential tables are available online for readers who wish to examine the dataset in greater depth.

### Statistical Analysis

Linear plots were used to track the number of patients diagnosed with a muscle wasting-associated disease by year. Data collection, organization, and graph creation were performed using Microsoft Excel (Microsoft Office Home and Student, 2019). Descriptive statistics were analyzed by first assessing for normal distribution of continuous variables with the Kolmogorov–Smirnov test. Normally distributed data were described as means and standard deviations, whereas non-normally distributed data were described with medians and interquartile ranges (IQR). Comparisons for the muscle wasting groups and patients who did not have muscle wasting diagnoses were performed with analysis of variance (ANOVA) tests for normally distributed data and Kruskal–Wallis tests for non-normally distributed data. Chi-squared tests were used to compare categorical data.

If adequate data were available for regression analysis by specific type of muscle wasting condition, then analysis was performed by specific condition (sarcopenia, protein malnutrition, or cachexia) to best determine the effect of the specific muscle wasting conditions on clinical outcomes. Linear regression was performed for continuous outcomes and binomial logistic regression was performed for categorical outcomes. All statistical analyses were performed with IBM SPSS Statistics (Version 29.0.2.0).

Inclusion criteria comprised all BCQP patients with available demographic, injury, and outcome data between 2000 and 2022. Patients with incomplete data were handled using pairwise deletion, consistent with BCQP registry practice. Diagnoses were identified by ICD-9/10 codes (Table A1 and Table A2). No matching was performed, as the analysis aimed to evaluate unadjusted population level associations across the entire cohort. Adjustment for multiple comparisons was not applied because each outcome represented a separate clinical endpoint.

Incomplete data were analyzed using pairwise deletion. Confounding variables were evaluated for potential inclusion in models using Pearson correlations. Variables were considered if correlation coefficients were |>0.1| and a Variance Inflation Factor < 10 to eliminate collinearity. Models were assessed for statistical significance with analysis of variance (ANOVA) for linear regression and Chi-Square Omnibus Test for binomial logistic regression. Statistical significance was set at an alpha of 0.05 for all tests.

## 3. Results

The BCQP represented a total number of 284,194 burn patients between 2000 and 2022. A total of 3284 (1.2%) of these patients fit the stated criteria for “muscle wasting” with a diagnosis of either sarcopenia, cachexia, or protein malnutrition. All patients with muscle wasting were diagnosed between the years of 2017 and 2022. Apparent absence of cases from 2013 to 2016 likely reflects limited adoption of muscle wasting ICD codes within the BCQP during the early ICD-10 transition period, rather than a true lack of such patients. Reporting frequency for cachexia and protein-malnutrition codes increased steadily after 2016, consistent with broader national reporting of these diagnostic codes.

Notably, there was a total increase per year in the diagnoses of muscle wasting (Figure 1), cachexia (Figure 2), and protein malnutrition (Figure 3) during the years 2017 to 2019, but these rates did not increase as a percentage of total annual patients enrolled in the BCQP. This classification was based on ICD codes for sarcopenia (n = 4, 0.001%), cachexia (n = 209, 0.07%), or protein malnutrition (n = 3215; 1.12%). A total of 144 patients had two or more concurrent muscle wasting diagnoses. Only four patients in the BCQP were diagnosed with sarcopenia—one each in 2019 and 2022, and two with unspecified diagnosis years. Comparisons of the sarcopenia patients were not statistically or clinically meaningful and are listed in Table A3 for reference but were excluded from further analysis.

Patients with muscle wasting were significantly older (sarcopenia: 56.5 years [IQR: 35.5–70.8], cachexia: 63.0 years [IQR: 52.0–72.0], protein malnutrition: 57.0 years [IQR: 41.0–68.0], Table 1) vs. non-muscle wasting patients (37.0 years [IQR: 19.0–56.0]; *p* < 0.001). Additionally, patients with muscle wasting conditions had larger TBSA burns (sarcopenia: 27.0% [IQR: 14.5–46.0%], cachexia: 4.0% [IQR 1.5–13.8%], protein malnutrition 9.0% [IQR: 2.0–24.9%]) compared to non-muscle wasting patients (2.0% [IQR: 0.0–6.0%]; *p* < 0.001). Furthermore, muscle wasting patients were more likely to be female (protein malnutrition 36.2% female vs. non-muscle wasting 33.8%, *p* = 0.005), more likely to be unemployed (cachexia: 87.5%, protein malnutrition: 67.2% vs. non-muscle wasting: 43.3%, *p* < 0.001], and more likely to live alone (cachexia: 39.6%, protein malnutrition: 34.5% vs. non-muscle wasting: 22.4%, *p* < 0.001). There were no significant differences between cohorts in terms of race, marital status, burn mechanism, insurance status, and rates of most medical comorbidities (Table A3).

Regression analysis with data collected from cachectic patients demonstrated that cachexia significantly predicted a higher likelihood of mortality (Odds Ratio (OR) 2.2 [95% Confidence Interval (CI): 1.3–3.7], *p* = 0.004, Table 2). In contrast, protein malnutrition was not associated with increased mortality (OR 1.12 [95% CI: 0.93–1.35], *p* = 0.239, Table 2). Meanwhile, muscle wasting conditions were significantly associated with decreased likelihoods of being discharged home to independent living (cachexia: OR 0.3 [95% CI: 0.2–0.5], protein malnutrition: OR 0.4 [95% CI: 0.4–0.5]; *p* < 0.0001 for both, Table 3).

Additionally, patients with muscle wasting conditions were at higher risk of developing sepsis (cachexia: OR 16.2 [95% CI: 10.8–24.3]; protein malnutrition: OR 16.2 [95% CI: 14.3–18.4], *p* < 0.0001 for both, Table 4) and wound infection/skin graft loss (cachexia: OR 2.6 [95% CI: 1.3–5.5], *p* = 0.01; protein malnutrition: OR 3.5 [95% CI: 2.9–4.2], *p* < 0.0001, Table 5). Patients with protein malnutrition had a significantly higher risk of requiring more skin debridement procedures compared to patients without muscle wasting (OR 16.8 [95% CI: 6.3–27.3], *p* = 0.002, Table 6), but this association was not evident in patients diagnosed with cachexia (OR 15.0 [95% CI: −25.7–55.7], *p* = 0.470, Table 6). Patients with cachexia and protein malnutrition were more likely to have longer hospital stays (cachexia: OR 11.2 [95% CI: 8.4–14.1], protein malnutrition (OR 13.4 [95% CI: 12.7–14.2]; both *p* < 0.0001) compared to non-muscle wasting patients (Table 7). Moreover, patients with muscle wasting conditions had increased risk of longer ICU stays and days requiring ventilator support that reached statistical significance (*p* < 0.0001 for all, see Table A4 and Table A5).

## 4. Discussion

In 2019, the American Burn Association (ABA) launched the BCQP, integrating data from the original National Burn Repository database and the Burn Quality Improvement Program into one entity, the largest database of burn patients to date [12]. With the updated version of the BCQP that included increased patient enrollment from 2017–2022, a more powerful statistical analysis was performed with this larger population to better understand how muscle wasting conditions may affect outcomes for burn patients. Increasing the number of patients included had the potential to reduce selection bias when analyzing the effects of preexisting muscle wasting conditions on burn outcomes. However, the proportion of patients diagnosed with muscle wasting conditions reduced from 2.6% to 1.2%, either representing a true decrease in the number of patients with these diagnoses or an extrapolation of the selection bias from the previous database. Understanding how preexisting muscle wasting influences infection risk, wound healing, ventilator dependence, and mortality can guide earlier nutritional and rehabilitative interventions in burn management.

Interestingly, there are several reasons why the lack of patients diagnosed with muscle wasting conditions in the BCQP does not reflect a true decrease in the incidence of these conditions. The percentage of patients in the latest BCQP aged 65 or older is 14%, where the baseline incidence of sarcopenia is estimated to be between 5–22% [11,13,14,15]. Therefore, the expected incidence of sarcopenia in the BCQP would be 0.7–3%; however, with only four patients being diagnosed with sarcopenia (<0.001%) in this database, this lower rate likely reflects lack of reporting of this diagnosis. Moreover, “sarcopenia” is a relatively new ICD-10-CM code proposed in 2016, and according to the definition, there are varying rates of incidence [16]. Moreover, the definition was updated in 2019 by the European Working Group on Sarcopenia in Older People (EWGSOP) [15,17]. Since this diagnosis is associated with a newer concept of muscle atrophy with aging, this condition may be under-recognized in the burn patient populations enrolled in the BCQP. Since elderly patients are disproportionately affected by major burns, increasing awareness of sarcopenia could help with more accurate assessment of its impact on outcomes that are particularly important in the care of elderly burn patients, such as mortality and discharge to home independent living [18,19,20].

An observed uptick in the diagnoses of cachexia and protein malnutrition during 2017 to 2019 may reflect the adoption of more structured approaches to malnutrition screening, diagnosis, and documentation. For example, the Malnutrition Quality Improvement Initiative (MQii), a national effort led by the Academy of Nutrition and Dietetics and Avalere Health, disseminated quality measures and best practices aimed at improving the recognition and management of malnutrition, especially in hospitalized and older adult populations [21]. This initiative encouraged clinical assessments, such as reduced muscle mass and functional decline, with the diagnosis of malnutrition, thereby potentially increasing the number of patients diagnosed with this condition. Additionally, efforts to integrate malnutrition risk identification into electronic health records (EHRs), Centers for Medicare and Medicaid Services quality reporting programs, and transitional care models likely increased clinician awareness and coding accuracy [22]. As noted in a 2018 national dialogue hosted by the Malnutrition Quality Collaborative, malnutrition was historically under-recognized during transitions across care settings, despite its strong association with increased healthcare costs, complications, and mortality [23]. This meeting spurred actionable recommendations, such as adopting standardized malnutrition terminology in EHRs, educating clinicians on functional aspects of nutrition, and aligning reimbursement models with nutrition care delivery, which may have accelerated institutional efforts to identify and document malnutrition more rigorously.

Despite the raw increase in the number of patients diagnosed with cachexia and protein malnutrition after 2017, this analysis demonstrated a stable rate of these diagnoses as a percentage of annual enrollment since 2017. Moreover, there was an overall decrease in the number of patients with muscle wasting conditions as a total of the BCQP. As previously noted, the BCQP database does not require patients with less than 10% TBSA to provide a complete report of preexisting ICD diagnoses, compounding selection bias from those who are enrolled that have smaller TBSAs [1]. Additionally, it is likely that the rate of diagnoses of cachexia and protein malnutrition was still proportionally lower than the overall enrollment in burn patients to the BCQP. Therefore, despite the improvement in reporting of these conditions with increased awareness, the overall proportion of patients with these muscle wasting conditions was lower in this newer dataset compared to the previous dataset. This decrease also coincides with recommendations to start enteral nutrition within the first 4–6 h post-injury and a move toward volume-based feeding, which may also influence diagnosis rates [24,25].

Despite the low incidence of muscle wasting conditions among patients in the BCQP, this analysis examines the distinctions between two types of muscle wasting: cachexia and protein malnutrition. These two subtypes negatively contributed to a variety of clinical outcomes. Unlike previous analysis, preexisting cachexia was a significant mortality predictor in burn patients (OR 2.2, *p* = 0.004). With increased power in this newer analysis, these results suggest that cachexia may present a more immediate risk to survival rates in patients with large burns, perhaps explained by previous findings of impaired immune responses and wound healing [8]. However, this effect was not present with protein malnutrition. At present it is unclear whether there may be additional impacts of cachexia from underlying illnesses that predispose patients to mortality that are not present in patients who have sarcopenia or protein malnutrition. Additional enrollment is needed to adequately power the analysis to demonstrate the true effects of cachexia as well as sarcopenia and protein malnutrition.

Conversely, both cachexia and protein malnutrition had statistically significant impacts on most other clinical outcomes, such as likelihood to discharge home to independent living, sepsis, wound infection, and length of stay. When comparing the OR of outcomes between protein malnutrition and cachexia, there was a more pronounced effect on almost all outcomes from protein malnutrition, rather than cachexia. Sepsis is a major complication in burn patients, and cachexia and protein malnutrition were associated with an increase in sepsis development. Previous studies have also noted the persistence of muscle catabolism after severe burns and the patients’ resulting inability to fight infections, leading to complications such as sepsis [26]. In addition, patients with cachexia and protein malnutrition spent significantly more time in the hospital and ICU and on ventilator support compared to those without these conditions, similar to observations in our previous study [1].

These differences are consistent with the underlying biology of muscle wasting syndromes. Protein-calorie malnutrition reflects acute substrate deficiency in the setting of the burn hypermetabolic response, which intensifies catabolism and impairs wound healing, immune function, and respiratory muscle performance, mechanisms that could plausibly lengthen ICU and hospital stay and increase ventilator days. Cachexia, by contrast, is a chronic, disease-related inflammatory state with progressive muscle loss and altered energy metabolism; its association with mortality can be strong even when effects on hospital or ICU utilization appear smaller. Differences in inflammatory milieu, substrate availability, and respiratory muscle function therefore provide a coherent physiologic rationale for the effect patterns observed here [27]. Future studies incorporating functional and biochemical markers (e.g., handgrip strength, DXA/ultrasound, serum proteins, inflammatory cytokines) are needed to test these mechanisms directly.

Importantly, the BCQP does not report many metrics assessing the severity of preexisting muscle wasting. Standard clinical assessments, such as imaging (e.g., DXA, ultrasound, or MRI) or proxy measures from laboratory values such as creatinine, have not been widely adopted by burn care providers. Additionally, the ICD codes that the patients are diagnosed with do not have an associated timing of diagnosis, making it difficult for the user to distinguish whether or not a given muscle wasting condition is preexisting or acquired during the acute stay of his or her burn care. For some terms, such as “sarcopenia” or “cachexia,” there is an association of chronicity which implies that the conditions are preexisting; however, with “protein malnutrition,” it is unclear if this condition is due to effects from burn hypermetabolism or preexisting factors unrelated to the burn injury. One way to improve this distinction would be to introduce burn hypermetabolism as an ICD code and use specific criteria to define the severity of both the burn hypermetabolism and pre-admission muscle wasting.

More thorough assessment of muscle wasting disease in burn patients would likely result in more accurately diagnosis of underlying causes of muscle loss, whether due to sarcopenia, cachexia, protein malnutrition, or other types of muscle loss, such as hypermetabolism. Additional study with integration of more of these clinical measures, particularly with objective measurements of muscle strength, would be needed to more accurately assess the true impact of these conditions on healing in burn patients. These data would provide additional information for distinguishing the physiological differences in these types of muscle wasting, which could better establish which aspects of muscle wasting predispose to the greatest morbidity. Ultimately, improving the understanding of how muscle wasting conditions affect burn recovery is one way to optimize interventions to reduce muscle loss and improve clinical outcomes in these patients.

### Limitations

This study has several limitations. The BCQP does not specify the timing or chronicity of muscle wasting diagnoses, preventing differentiation between preexisting and burn-acquired malnutrition. Underreporting and inconsistent coding likely reduced the apparent prevalence of muscle wasting conditions, particularly before 2017. Residual confounding is possible, as regression models lacked matching or comprehensive adjustment for comorbidities. Objective measures of muscle mass or function (such as DXA, ultrasound, or handgrip strength) are unavailable in the BCQP, limiting physiologic interpretation. Finally, excluding pre-2017 patients with potential unrecorded muscle wasting may have introduced selection bias. These factors should be considered when interpreting the findings.

## 5. Conclusions

In the BCQP, the largest database of burn patients available, patients with muscle wasting conditions, such as sarcopenia, cachexia, and protein malnutrition, were analyzed to better understand the reporting of these conditions and their impacts on burn outcomes. The overall proportion of patients diagnosed with muscle wasting conditions decreased with the newer database (1.2% vs. 2.6%). While the incidence of cachexia and protein malnutrition reporting increased in 2017, the prevalence stabilized in ensuing years at a rate that is lower than what would be expected from baseline population values, while sarcopenia remains essentially unreported. Despite the low prevalence of these conditions, cachexia and/or protein malnutrition were associated with poorer outcomes for burn patients including mortality, infection rate, and hospital stay length, suggesting that these conditions are critical to optimizing care for burn patients.

## Figures and Tables

**Figure 1 ebj-06-00061-f001:**
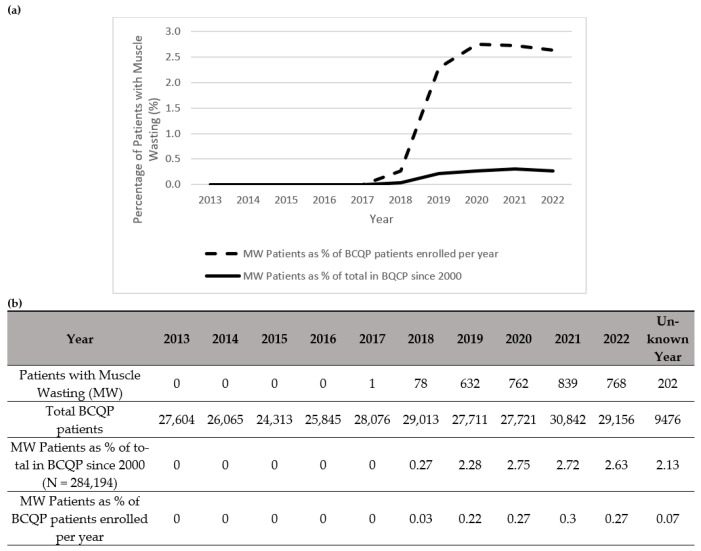
Summary of patients in the Burn Care Quality Platform (BCQP) diagnosed with muscle wasting (**a**) graphically between 2013 and 2022 and (**b**) by year, expressed as a percentage of the total number of BCQP patients enrolled since 2000 and for each individual year. For the purposes of this analysis, “muscle wasting” is comprised of International Classification Disease (ICD) codes associated with “sarcopenia,” “cachexia,” and “protein malnutrition” (See Table A1).

**Figure 2 ebj-06-00061-f002:**
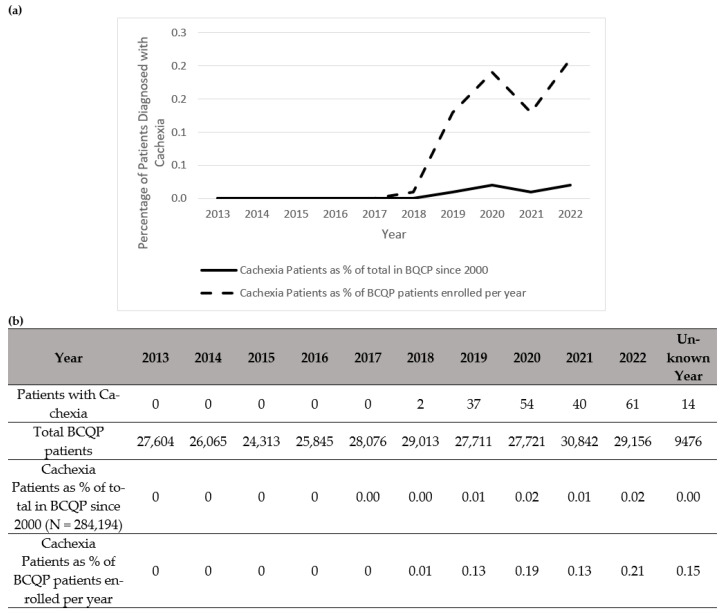
Summary of patients in the Burn Care Quality Platform (BCQP) diagnosed with cachexia (**a**) graphically between 2013 and 2022 and (**b**) by year, expressed as a percentage of the total number of BCQP patients enrolled since 2000 and for each individual year.

**Figure 3 ebj-06-00061-f003:**
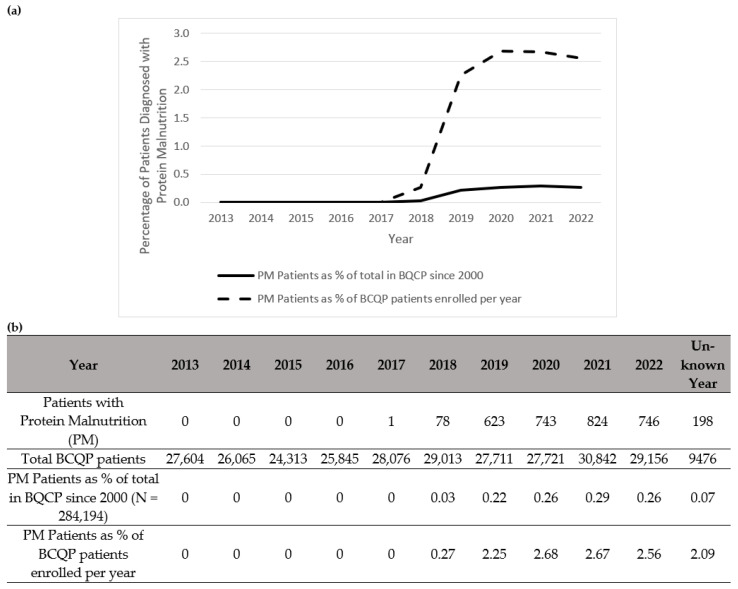
Summary of patients in the Burn Care Quality Platform (BCQP) diagnosed with protein malnutrition (**a**) graphically between 2013 and 2022 and (**b**) by year, expressed as a percentage of the total number of BCQP patients enrolled since 2000 and for each individual year.

**Table 1 ebj-06-00061-t001:** Abbreviated comparison between patients diagnosed with different muscle wasting conditions in the Burn Care Quality Platform (BCQP) based on demographics, burn characteristics, and outcomes from 2000–2022. BCQP ICD codes do not have an associated timing of diagnosis.

Variable	Diagnosis of Sarcopenia (N = 4)	Diagnosis of Cachexia (N = 209)	Diagnosis of Protein Malnutrition (N = 3215)	No Diagnosis of Muscle Wasting (N = 284,194)	*p*-Value
**Demographics**					
Age	56.5 (35.5–70.75)	63 (52–72)	57 (41–68)	37 (19–56)	<0.001
Total Body Surface Area (%)	27 (14.5–46)	4 (1.5–13.8)	9 (2–24.875)	2 (0–6)	<0.001
% Female	(0/4) 100%	(71/209) 33.8%	(1162/3214) 36.2%	(95,641/282,949) 33.8%	0.005
**Living Characteristics**					
% Unemployed Status	(1/1) 100%	(56/64) 87.5%	(415/618) 67.2%	(31,825/73,526) 43.3%	<0.001
% Living Alone Status	(1/4) 25%	(76/192) 39.6%	(1013/2937) 34.5%	(39,310/181,032) 22.4%	<0.001
**Burn Characteristics**					
% Inhalation Injury Present	(1/2) 50%	(22/193) 11.4%	(381/2659) 14.3%	(19,985/275,626) 7.3%	<0.001
**Insurance Characteristics**					
% MediCARE Insured	(2/4) 50%	(104/204) 51.0%	(1269/3169) 40.0%	(43,701/269,964) 16.2%	<0.001
**Outcomes**					
Number of Total Days Inpatient	42 (29–57.5)	14 (5–35)	19 (8–37)	4 (1–10)	<0.001
Number of Total Days in Intensive Care	36.5 (19.25–53.75)	7 (2–24.5)	10 (3–27)	1 (0–4)	<0.001
Number of Total Days on Ventilator	2 (1–2)	1 (0–5)	3 (0–12)	0 (0–0)	<0.001

A table outlining all demographics and patient outcomes can be found in Table A3.

**Table 2 ebj-06-00061-t002:** Likelihood of mortality in Burn Care Quality Platform (BCQP) patients with cachexia or protein malnutrition from 2000–2022.

Variable	B	S.E.	Wald	OR (95% C.I.)	*p*-Value
Diagnosis of Cachexia	0.782	0.274	8.139	2.186 (1.277–3.741)	0.004
Male Sex	−0.084	0.033	6.675	0.919 (0.863–0.980)	0.010
Age in Years	0.053	0.001	3880.512	1.055 (1.053–1.057)	<0.0001
TBSA (%)	0.065	0.001	7784.970	1.067 (1.066–1.069)	<0.0001
Presence of Inhalation Injury	2.044	0.033	3889.634	7.719 (7.239–8.231)	<0.0001
History of Major Cardiovascular Disease Comorbidity	0.922	0.049	355.991	2.514 (2.284–2.767)	<0.0001
Constant	−7.360	0.063	13,691.431	0.001	<0.0001
Diagnosis of Protein Malnutrition	0.114	0.097	1.389	1.121 (0.927–1.354)	0.239
Male Sex	−0.084	0.033	6.743	0.919 (0.862–0.979)	0.009
Age in Years	0.053	0.001	3887.400	1.055 (1.053–1.057)	<0.0001
TBSA (%)	0.065	0.001	7735.169	1.067 (1.066–1.069)	<0.0001
Presence of Inhalation Injury	2.044	0.033	3888.972	7.718 (7.238–8.230)	<0.0001
History of Major Cardiovascular Disease Comorbidity	0.924	0.049	357.545	2.519 (2.289–2.772)	<0.0001
Constant	−7.476	0.116	4164.155	0.001	<0.0001

C.I.: Confidence Interval. S.E.: Standard Error. TBSA: Total Body Surface Area.

**Table 3 ebj-06-00061-t003:** Likelihood of discharge home to independent living vs. other location or circumstances in Burn Care Quality Platform (BCQP) patients with cachexia or protein malnutrition from 2000–2022.

Variable	B	S.E.	Wald	OR (95% C.I.)	*p*-Value
Diagnosis of Cachexia	−1.200	0.208	33.212	0.301 (0.200–0.453)	<0.0001
Age in Years	−0.029	0.000	5176.231	0.972 (0.971–0.973)	<0.0001
TBSA (%)	−0.049	0.001	4841.270	0.952 (0.951–0.954)	<0.0001
Living Alone Status	−0.488	0.016	932.504	0.614 (0.595–0.633)	<0.0001
History of Major Cardiovascular Disease Comorbidity	−0.660	0.035	355.697	0.517 (0.482–0.553)	<0.0001
Medicare Insurance Status	−0.597	0.020	882.827	0.551 (0.529–0.573)	<0.0001
Constant	2.942	0.018	26,384.438	18.950	<0.0001
Diagnosis of Protein Malnutrition	−0.835	0.107	61.440	0.434	<0.0001
Age in Years	−0.027	0.001	818.130	0.973 (0.971–0.975)	<0.0001
TBSA (%)	−0.048	0.002	991.091	0.953 (0.950–0.956)	<0.0001
Presence of Inhalation Injury	−1.013	0.072	196.983	0.363 (0.315–0.418)	<0.0001
Unemployed Status	−0.635	0.036	305.057	0.530 (0.493–0.569)	<0.0001
Living Alone Status	−0.404	0.036	126.451	0.668 (0.623–0.717)	<0.0001
History of Major Cardiovascular Disease Comorbidity	−0.525	0.083	40.322	0.592 (0.503–0.696)	<0.0001
Medicare Insurance Status	−0.250	0.053	21.864	0.779 (0.702–0.865)	<0.0001
Constant	3.184	0.047	4656.570	24.153	<0.0001

C.I.: Confidence Interval. S.E.: Standard Error. TBSA: Total Body Surface Area.

**Table 4 ebj-06-00061-t004:** Likelihood of developing sepsis during acute burn treatment in Burn Care Quality Platform (BCQP) patients with cachexia or protein malnutrition from 2000–2022.

Variable	B	S.E.	Wald	OR (95% C.I.)	*p*-Value
Diagnosis of Cachexia	2.785	0.206	183.105	16.201 (10.823–24.252)	<0.0001
Age in Years	0.015	0.001	258.504	1.016 (1.014–1.017)	<0.0001
TBSA (%)	0.039	0.001	2180.751	1.040 (1.038–1.042)	<0.0001
Presence of Inhalation Injury	0.933	0.053	310.823	2.543 (2.292–2.821)	<0.0001
Year of Injury	0.145	0.007	403.140	1.156 (1.140–1.172)	<0.0001
History of Diabetes Mellitus	0.551	0.054	103.057	1.735 (1.560–1.929)	<0.0001
Constant	−298.048	14.564	418.816	0.000	<0.0001
Diagnosis of Protein Malnutrition	2.788	0.064	1905.593	16.242 (14.332–18.408)	<0.0001
Age in Years	0.014	0.001	192.531	1.014 (1.012–1.016)	<0.0001
TBSA (%)	0.038	0.001	1907.161	1.039 (1.037–1.040)	<0.0001
Presence of Inhalation Injury	0.944	0.054	305.466	2.570 (2.312–2.857)	<0.0001
Year of Injury	0.104	0.008	189.493	1.109 (1.093–1.126)	<0.0001
History of Diabetes Mellitus	0.522	0.056	87.053	1.685 (1.510–1.880)	<0.0001
Constant	−214.648	15.177	200.020	0.000	<0.0001

C.I.: Confidence Interval. S.E.: Standard Error. TBSA: Total Body Surface Area.

**Table 5 ebj-06-00061-t005:** Likelihood of developing a wound infection and/or graft loss during acute burn treatment in Burn Care Quality Platform (BCQP) patients with cachexia or protein malnutrition from 2000–2022.

Variable	B	S.E.	Wald	OR (95% C.I.)	*p*-Value
Diagnosis of Cachexia	0.968	0.377	6.591	2.632 (1.257–5.509)	0.010
Age in Years	0.012	0.001	111.605	1.012 (1.010–1.015)	<0.0001
TBSA (%)	0.031	0.001	607.954	1.032 (1.029–1.034)	<0.0001
Presence of Inhalation Injury	−0.184	0.090	4.225	0.832 (0.698–0.991)	0.040
History of Diabetes Mellitus	0.153	0.071	4.570	1.165 (1.013–1.340)	0.033
Number of Days Between Injury and First Procedure	0.008	0.001	78.062	1.009 (1.007–1.010)	<0.0001
Constant	−4.748	0.059	6511.119	0.009	<0.0001
Diagnosis of Protein Malnutrition	1.252	0.092	183.553	3.498 (2.918–4.192)	<0.0001
Age in Years	0.012	0.001	96.386	1.012 (1.009–1.014)	<0.0001
TBSA (%)	0.029	0.001	513.212	1.030 (1.027–1.032)	<0.0001
Presence of Inhalation Injury	−0.190	0.089	4.515	0.827 (0.694–0.985)	0.034
History of Diabetes Mellitus	0.131	0.072	3.368	1.140 (0.991–1.312)	0.066
Number of Days between Injury and First Procedure	0.008	0.001	73.256	1.008 (1.006–1.010)	<0.0001
Constant	−4.745	0.059	6475.274	0.009	<0.0001

C.I.: Confidence Interval. S.E.: Standard Error. TBSA: Total Body Surface Area.

**Table 6 ebj-06-00061-t006:** Likelihood of skin debridement procedures required during acute burn care for Burn Care Quality Platform (BCQP) patients diagnosed with cachexia or protein malnutrition from 2000–2022.

Variable	B (95% C.I.)	S.E.	β	t	*p*-Value
Diagnosis of Cachexia	14.972 (−25.727–55.672)	20.732	0.025	0.722	0.470
Age in Years	−0.155 (−0.213–0.098)	0.029	−0.230	−5.335	<0.0001
TBSA (%)	0.234 (0.130–0.338)	0.053	0.160	4.416	<0.0001
Presence of Inhalation Injury	5.171 (0.767–9.576)	2.244	0.084	2.305	0.021
Year of Injury	0.428 (0.049–0.807)	0.193	0.078	2.216	0.027
Medicare Insurance	3.955 (−0.124–8.035)	2.078	0.092	1.903	0.057
Other Type of Insurance	1.599 (−4.053–7.252)	2.879	0.020	0.555	0.579
Private Insurance	4.840 (2.052–7.628)	1.420	0.144	3.408	0.001
Self-Pay/Charity Care	2.436 (−1.173–6.046)	1.839	0.052	1.325	0.186
Constant	−847.261 (−1612.581–81.941)	389.844		−2.173	0.030
Diagnosis of Protein Malnutrition	16.782 (6.297–27.268)	5.341	0.111	3.142	0.002
Age in Years	0.220 (0.117–0.324)	0.029	−0.234	−5.468	<0.0001
TBSA (%)	0.220 (0.117–0.324)	0.053	0.151	4.176	<0.0001
Presence of Inhalation Injury	5.192 (0.815–9.570)	2.230	0.085	2.329	0.020
Year of Injury	0.374 (−0.005–0.752)	0.193	0.068	1.937	0.053
Medicare Insurance	3.805 (−0.250–7.861)	2.066	0.088	1.842	0.066
Other Type of Insurance	1.598 (−4.020–7.216)	2.862	0.020	0.558	0.577
Private Insurance	4.907 (2.136–7.678)	1.411	0.146	3.477	0.001
Self-Pay/Charity Care	2.429 (−1.158–6.016)	1.827	0.052	1.330	0.184
Constant	−737.125 (−1500.984–26.733)	389.099		−1.894	0.059

C.I.: Confidence Interval. S.E.: Standard Error. TBSA: Total Body Surface Area.

**Table 7 ebj-06-00061-t007:** Likelihood of increased total length of inpatient stay for Burn Care Quality Platform (BCQP) patients diagnosed with cachexia or protein malnutrition from 2000–2022.

Variable	B (95% C.I.)	S.E.	β	t	*p*-Value
Diagnosis of Cachexia	11.241 (8.349–14.133)	20.732	0.025	0.722	0.470
Male Sex	−0.390 (−0.555–0.225)	0.029	−0.230	−5.335	<0.0001
Age in Years	0.075 (0.072–0.078)	0.053	0.160	4.416	<0.0001
TBSA (%)	0.519 (0.512–0.526)	2.244	0.084	2.305	0.021
Presence of Inhalation Injury	2.409 (2.096–2.722)	0.193	0.078	2.216	0.027
Year of Injury	−0.022 (−0.049–0.005)	2.078	0.092	1.903	0.057
Homeless Status	2.649 (1.932–3.367)	2.879	0.020	0.555	0.579
Living in Own Home/Apartment	−1.853 (−2.447–1.259)	1.420	0.144	3.408	0.001
Constant	49.122 (−5.209–103.454)	27.721		1.772	0.076
Diagnosis of Protein Malnutrition	13.439 (12.693–14.186)	0.381	0.085	35.287	<0.0001
Male Sex	−0.359 (−0.523–0.194)	0.084	−0.010	−4.269	<0.0001
Age in Years	0.072 (0.068–0.075)	0.002	0.102	41.953	<0.0001
TBSA (%)	0.509 (0.501–0.516)	0.004	0.337	134.942	<0.0001
Presence of Inhalation Injury	2.419 (2.108–2.731)	0.159	0.038	15.221	<0.0001
Year of Injury	−0.066 (−0.093–0.039)	0.014	−0.012	−4.786	<0.0001
Homeless Status	2.603 (1.889–3.317)	0.364	0.029	7.142	<0.0001
Constant	137.785 (83.436–192.134)	27.729		4.969	<0.0001

C.I.: Confidence Interval. S.E.: Standard Error. TBSA: Total Body Surface Area.

## Data Availability

Restrictions apply to the availability of Burn Care Quality Platform (BCQP) data. Data were obtained from the American Burn Association (ABA) and are available at https://ameriburn.org/quality-care/burn-care-quality-platform-bcqp-registry/bcqp-full-registry/ (accessed on 30 March 2025) with the permission of the ABA. Further inquiries can be directed to the corresponding author.

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
