# Peer review of "The Role of Previous History of Muscle Wasting in Burn Outcomes—A Burn Care Quality Platform Study"

_2673-1991, 2025, doi:10.3390/ebj6040061_

Round 1
Reviewer 1 Report
Comments and Suggestions for Authors
Thank you for the opportunity to review, "The Role of Previous History of Muscle Wasting in Burn Outcomes – A Burn Care Quality Platform Study". It is well-known that major burns lead to lean body muscle loss. I commend the authors for trying to elucidate the risk that pre-burn muscle wasting imparts on burn patients' clinical outcomes.
I have a few recommendations and request clarity on the paper:
- Figures 1-3 are difficult to read and interpret in their current form, specifically the row "Total BCQP patients". I am not sure how much this adds to the paper and would recommend either reformatting the tables/figures or condensing them into a single figure, such as a table graph with columns for the 3 MW diagnoses, perhaps as percentage of BCQP patients enrolled yearly.
- You include BCQP patients from 2000-2022, but then only report on patients with MW diagnoses from 2017-2022. Are you suggesting that no patients prior to 2017 were diagnosed with a MW ICD code? This contradicts data that your group previously published in JBCR which shows patients diagnosed with MW dating back as early as 2013. Please explain why patients you excluded patients with MW diagnoses prior to 2017 (your numerator, as it were) but compared them to all BCQP patients over the study period (the denominator).
- If no diagnosis of MW was ever made before 2013, why include BCQP patients prior to that date? You state clearly that MW is under-diagnosed, which means patients with MW that were not coded as such are being included in the no MW group.
- You mention that patients diagnosed with protein malnutrition and cachexia had more time in the hospital, ICU and on ventilatory support, and these effects were more pronounced in the protein malnutrition group compared to cachexia (lines 311-316). Your explanation afterwards (316-322) does not make sense to me. You rightly say there is a higher diagnosis of protein malnutrition, which could influence the results, but then suggest patients with cachexia have more "compensations" (i.e. functional reserve?) compared to patients with protein malnutrition, but do not provide any supporting citations for that assertion. Please explain your reasoning.
- Please list limitations of the study more explicitly. You imply them in your Discussions but state them more generally (e.g. not knowing chronicity of MW diagnosis, under-reporting).
Author Response
We greatly appreciate the manuscript comments from Reviewer 1. Your feedback helped us clarify several critical points and greatly strengthened the manuscript. Below, we have specifically addressed each of your comments:
Comment 1: Figures 1-3 are difficult to read and interpret in their current form, specifically the row "Total BCQP patients". I am not sure how much this adds to the paper and would recommend either reformatting the tables/figures or condensing them into a single figure, such as a table graph with columns for the 3 MW diagnoses, perhaps as percentage of BCQP patients enrolled yearly.
Thank you for your observation and feedback. We have formatted the tables to remove extra information and decreased the font size for some rows to decrease hyphenation and improve spacing between values and readability. Please kindly refer to the revised manuscript, which is attached.
Comment 2: You include BCQP patients from 2000-2022, but then only report on patients with MW diagnoses from 2017-2022. Are you suggesting that no patients prior to 2017 were diagnosed with a MW ICD code? This contradicts data that your group previously published in JBCR which shows patients diagnosed with MW dating back as early as 2013. Please explain why patients you excluded patients with MW diagnoses prior to 2017 (your numerator, as it were) but compared them to all BCQP patients over the study period (the denominator).
Thank you for this observation; this point needs to be clarified in the manuscript. ICD-10-CM codes were nationally implemented in 2015, and the specific code for sarcopenia (M62.84) was introduced in 2016. Documentation of sarcopenia and cachexia (R64) appears in later BCQP records. Because earlier years showed minimal or inconsistent coding for these diagnoses, we limited muscle-wasting analyses to 2017–2022 while retaining the full 2000–2022 BCQP cohort for context. This is reflected in lines 90–95 in the revised manuscript.
Comment 3: If no diagnosis of MW was ever made before 2013, why include BCQP patients prior to that date? You state clearly that MW is under-diagnosed, which means patients with MW that were not coded as such are being included in the no MW group.
Thank you for bringing this to our attention. This point also needs to be addressed. Years prior to 2017 demonstrated minimal or inconsistent coding for cachexia and sarcopenia diagnoses. Because of this, we limited muscle-wasting analyses to 2017–2022 while retaining the full 2000–2022 BCQP cohort for context. Some patients with undiagnosed or miscoded muscle-wasting conditions before 2017 were likely categorized in the no muscle wasting group, reflecting under-recognition of these diagnoses during earlier registry years. We have included this explanation in the revised manuscript (lines 93-97).
Comment 4: You mention that patients diagnosed with protein malnutrition and cachexia had more time in the hospital, ICU and on ventilatory support, and these effects were more pronounced in the protein malnutrition group compared to cachexia (lines 311-316). Your explanation afterwards (316-322) does not make sense to me. You rightly say there is a higher diagnosis of protein malnutrition, which could influence the results, but then suggest patients with cachexia have more "compensations" (i.e. functional reserve?) compared to patients with protein malnutrition, but do not provide any supporting citations for that assertion. Please explain your reasoning.
Thank you for observing the lack of clarity and supporting references on this point. We have provided a more detailed description contrasting the differences between protein malnutrition and cachexia and added a reference citing the "[d]ifferences in inflammatory milieu, substrate availability, and respiratory muscle function" that provide a physiologic rationale for the effect patterns observed in the study. (Lines 352–363 in revised manuscript)
Comment 5: Please list limitations of the study more explicitly. You imply them in your Discussions but state them more generally (e.g. not knowing chronicity of MW diagnosis, under-reporting).
Thank you for this feedback. We have included a more general outline of the study limitations and the end of the revised manuscript, from lines 389 to 398.
Reviewer 2 Report
Comments and Suggestions for Authors
General:
This is a large database study between 2000 – 2022 of over a quarter of a million burn patients. The report is clearly written and reads well. I question the need for all the data to be published; particularly in relation to the appendices. Could this not be made available in online research databases instead?
Abstract:
This is well written.
Introduction:
The types of muscle wasting are well explained.
Results:
In Figure 1b, the “Total BCQP patients row is unclear as there are no gaps between data. Similarly in 2b and 3b
Author Response
We greatly appreciate the manuscript comments from Reviewer 2. Your feedback helped us clarify our data presentation and strengthen the manuscript. Below, we have addressed each of your comments:
Comment 1. I question the need for all the data to be published; particularly in relation to the appendices. Could this not be made available in online research databases instead?
Thank you for this feedback. We moved some of the data that is less critical to the results of the study online to reduce the size of the tables provided in text and as supplementary information. This point is addressed in lines 105–110 of the revised manuscript, which is attached.
Comment 2: In Figure 1b, the “Total BCQP patients row is unclear as there are no gaps between data. Similarly in 2b and 3b.
We have changed the formatting in the tables to create space between the values in Figures 1b, 2b and 3b. Please note these changes in the revised manuscript.
Reviewer 3 Report
Comments and Suggestions for Authors
Dear Authors,
Thank you for the opportunity to review the manuscript “The Role of Previous History of Muscle Wasting in Burn Outcomes – A Burn Care Quality Platform Study.” After careful review, I have concluded that it should be rejected. The paper discusses an important aspect of post-burn pathophysiology. However, the study lacks the scientific rigor, depth, and novelty needed to be truly informative for the scientific community.
Just at the beginning of this year, the study “The Influence of Muscle Wasting on Patient Outcomes among Burn Patients: A Burn Care Quality Platform Study” was published in the Journal of Burn Care and Research (PMID: 39441971), covering data from burn patients between 2000 and 2018. In the current study, you aimed to contribute to the literature by examining updated data from 2017 to 2022 without including the previously available data. This significantly limits the power and ability to draw meaningful conclusions, as hinted at in the discussion.
The manuscript is also poorly prepared. The clinical relevance of this study is not clearly outlined. The methods section lacks detailed information on relevant aspects such as inclusion and exclusion criteria, timing of diagnosis, and statistical methods—including adjustment for multiple comparisons and the absence of matching. The results section is underdeveloped. Figures 1-3 are misleading because they show a timeline from 2013 to 2022 without indicating the incidences for the reported outcomes between 2013 and 2017 (incidence = 0). Furthermore, the results section does not cover all relevant information presented in the tables, such as multiple sub-regression analyses that are not further discussed. The discussion is convoluted and does not coherently build on the reported results.
Author Response
We greatly appreciate the manuscript comments from Reviewer 3. Your feedback helped us clarify several critical points and greatly strengthened the manuscript. Below, we have specifically addressed each of your comments:
Comment 1: Just at the beginning of this year, the study “The Influence of Muscle Wasting on Patient Outcomes among Burn Patients: A Burn Care Quality Platform Study” was published in the Journal of Burn Care and Research (PMID: 39441971), covering data from burn patients between 2000 and 2018. In the current study, you aimed to contribute to the literature by examining updated data from 2017 to 2022 without including the previously available data. This significantly limits the power and ability to draw meaningful conclusions, as hinted at in the discussion.
Thank you for your observations and feedback. While ICD-10-CM codes were nationally implemented in 2015, the specific code for sarcopenia (M62.84) was introduced in 2016, and documentation of sarcopenia and cachexia (R64) appears in later BCQP records. Our prior analysis of BCQP data (2000–2017) demonstrated that patients with muscle wasting had prolonged hospitalization and adverse outcomes. Building on that work, we extended the analysis through 2022 to assess whether reporting and outcomes had changed. Where applicable, we noted where our current analysis differed from or supported our original analysis to provide context and potential explanations for our observations. Please refer to lines 22–24, 90–92, 294–296, 315–316, 331–334, 340–342 and 348–351 of the revised manuscript, which is attached.
Comment 2: The clinical relevance of this study is not clearly outlined.
Thank you for this feedback. In the Discussion, we clearly outlined the clinical relevance of our study in lines 274-277: Understanding how preexisting muscle wasting influences infection risk, wound healing, ventilator dependence, and mortality can guide earlier nutritional and rehabilitative interventions in burn management.
Comment 3: The methods section lacks detailed information on relevant aspects such as inclusion and exclusion criteria, timing of diagnosis, and statistical methods—including adjustment for multiple comparisons and the absence of matching.
Thank you for recognizing this omission. We have clarified this point in lines 128–134 in the Materials and Methods section:
Inclusion criteria comprised all BCQP patients with available demographic, injury, and outcome data between 2000 and 2022. Patients with incomplete data were handled using pairwise deletion, consistent with BCQP registry practice. Diagnoses were identified by ICD-9/10 codes (Tables A1–A2). No matching was performed, as the analysis aimed to evaluate unadjusted population level associations across the entire cohort. Adjustment for multiple comparisons was not applied because each outcome represented a separate clinical endpoint.
Comment 4: Figures 1-3 are misleading because they show a timeline from 2013 to 2022 without indicating the incidences for the reported outcomes between 2013 and 2017 (incidence = 0).
Thank you for your feedback. We have clarified this point by adding the following explanation in the Results section (lines 146–150): Apparent absence of cases from 2013 to 2016 likely reflects limited adoption of muscle-wasting ICD codes within the BCQP during the early ICD-10 transition period, rather than a true lack of such patients. Reporting frequency for cachexia and protein-malnutrition codes increased steadily after 2016, consistent with broader national reporting of these diagnostic codes.
Comment 5: Furthermore, the results section does not cover all relevant information presented in the tables, such as multiple sub-regression analyses that are not further discussed.
Thank you for this feedback. The BCQP analysis provided us a wealth of information regarding the effects of muscle wasting on burn outcomes, and we did our best to highlight the results we found most clinically significant while providing all results to the reader. We provided an explanation for our rationale in the Materials and Methods section in lines 128–134:
To enhance transparency and allow reproducibility of our methodology, extended variable definitions, coding criteria, and supplementary analyses are provided in the Appendix. Only material essential for interpretation of the primary and secondary outcomes has been retained within the manuscript, while additional non-essential tables are available online for readers who wish to examine the dataset in greater depth.
Comment 6: The discussion is convoluted and does not coherently build on the reported results.
Thank you for your feedback on the Discussion section of the manuscript. We added several sections to the Discussion section to improve readability, flow and understanding of the manuscript and study significance. Please refer to lines 274–277, 352-363 and 388–398 of the revised manuscript.
Reviewer 4 Report
Comments and Suggestions for Authors
A very interesting topic, often overlooked in patients with major burns, which has a significant impact on outcomes and mortality. I suggest the following revisions to improve the manuscript and make it ready for publication:
- In the abstract, the sentence from paragraph 22–24 should be revised, as it refers to previous data from 2000–2017, while your study includes data from 2000–2022. This may confuse readers, so please correct it.
- Figure 1: The chart is excellent, but the accompanying table is poorly formatted — the numbers are difficult to follow due to lack of spacing. Please adjust the table by adding dividing lines. The same applies to Figures 2 and 3.
- Tables 2–7 are rather unclear and difficult to read, while the text itself is very clear. Please revise them to improve readability. In contrast, Table 1 is fully clear. Overall, I would suggest improving the presentation of results — tables in the appendix may remain detailed, but those within the text should be concise and easy to interpret.
- From a clinical standpoint, distinguishing between all these muscle wasting conditions — cachexia, sarcopenia, and protein deficiency — is particularly challenging, especially in burn patients. It would therefore be useful to include in the introduction how each condition was diagnosed and which parameters or threshold values were used to differentiate them.
Author Response
We greatly appreciate the manuscript comments from Reviewer 4. Your feedback helped us clarify several critical points and greatly strengthened the manuscript. Below, we have specifically addressed each of your comments:
Comment 1: In the abstract, the sentence from paragraph 22–24 should be revised, as it refers to previous data from 2000–2017, while your study includes data from 2000–2022. This may confuse readers, so please correct it.
Thank you for your careful observation and feedback. We have corrected this sentence in the revised manuscript (lines 22–24), which is attached: A prior analysis of BCQP data (2000–2017) demonstrated that patients with muscle wasting had prolonged hospitalization and adverse outcomes. Building on that work, we extended the analysis through 2022 to assess whether reporting and outcomes had changed.
Comment 2: Figure 1: The chart is excellent, but the accompanying table is poorly formatted — the numbers are difficult to follow due to lack of spacing. Please adjust the table by adding dividing lines. The same applies to Figures 2 and 3.
Thank you for your feedback. We have formatted the accompanying tables for Figures 1 through 3 to include spaces between the values to improve readability.
Comment 3: Tables 2–7 are rather unclear and difficult to read, while the text itself is very clear. Please revise them to improve readability. In contrast, Table 1 is fully clear. Overall, I would suggest improving the presentation of results — tables in the appendix may remain detailed, but those within the text should be concise and easy to interpret.
Thank you for your observation and feedback. We modified the table text to reduce hyphenation improve readability. We also included text in lines 107-110 to inform readers that supplemental information is available that could not be included directly in the manuscript.
Comment 4. From a clinical standpoint, distinguishing between all these muscle wasting conditions — cachexia, sarcopenia, and protein deficiency — is particularly challenging, especially in burn patients. It would therefore be useful to include in the introduction how each condition was diagnosed and which parameters or threshold values were used to differentiate them.
This is an excellent point. We added lines 54-59 in the Materials and Methods section to describe diagnostic criteria: In this analysis, each condition was defined strictly by its ICD coding criteria (Table A1): protein malnutrition = E43–E46; cachexia = R64; sarcopenia = M62.50. No laboratory thresholds were applied because such parameters are not available in the BCQP. While clinical overlap is recognized, the use of ICD-based definitions provide a standardized and reproducible approach for distinguishing these conditions in large-scale datasets.
Round 2
Reviewer 1 Report
Comments and Suggestions for Authors
Thank you to the authors for thoughtful consideration of my comments. I am satisfied with the responses.
Author Response
Round 2, Comment 1: Thank you to the authors for thoughtful consideration of my comments. I am satisfied with the responses.
Response: Thank you for your comments and suggestions. Your feedback has greatly improved the readability and quality of our manuscript. Please note that we have added vertical lines to our tables to further enhance readability.
Reviewer 3 Report
Comments and Suggestions for Authors
Dear Authors,
thank you for revising the manuscript. However, I still recommend some substantial revisions.
Comment 1: Just at the beginning of this year, the study “The Influence of Muscle Wasting on Patient Outcomes among Burn Patients: A Burn Care Quality Platform Study” was published in the Journal of Burn Care and Research (PMID: 39441971), covering data from burn patients between 2000 and 2018. In the current study, you aimed to contribute to the literature by examining updated data from 2017 to 2022 without including the previously available data. This significantly limits the power and ability to draw meaningful conclusions, as hinted at in the discussion.
Thank you for your observations and feedback. While ICD-10-CM codes were nationally implemented in 2015, the specific code for sarcopenia (M62.84) was introduced in 2016, and documentation of sarcopenia and cachexia (R64) appears in later BCQP records. Our prior analysis of BCQP data (2000–2017) demonstrated that patients with muscle wasting had prolonged hospitalization and adverse outcomes. Building on that work, we extended the analysis through 2022 to assess whether reporting and outcomes had changed. Where applicable, we noted where our current analysis differed from or supported our original analysis to provide context and potential explanations for our observations. Please refer to lines 22–24, 90–92, 294–296, 315–316, 331–334, 340–342 and 348–351 of the revised manuscript, which is attached.
Thank you for your clarification. However, the manuscript would still benefit from clarifications. Figure 1 suggests that there was no reporting of any muscle wasting conditions prior to 2017. Can you explain that? In the methods section you mention that: “Some patients with undiagnosed or miscoded muscle-wasting conditions before 2017 were likely categorized in the no muscle wasting group, reflecting under- recognition of these diagnoses during earlier registry years. (lines 95-97)” However, in the discussion you state that: “Moreover, there was an overall decrease in the number of patients with muscle wasting conditions as a total of the BCQP. “ (lines 315-316)” The sentences contradict each other.
A suggestion would be to first report the prevalences of all muscle wasting conditions for each year, since the coding changed throughout your study period. Then, you could report the prevalences for the subcategories, beginning with e.g. the year when the distinct code was implemented. I would also suggest stating the population (no., years) that was used for these subcategories.
Comment 2: The clinical relevance of this study is not clearly outlined.
Thank you for this feedback. In the Discussion, we clearly outlined the clinical relevance of our study in lines 274-277: Understanding how preexisting muscle wasting influences infection risk, wound healing, ventilator dependence, and mortality can guide earlier nutritional and rehabilitative interventions in burn management.
I appreciate you adding that. Based on this, I would suggest strengthening your future outlook with more actionable recommendations for clinical practice.
Comment 3: The methods section lacks detailed information on relevant aspects such as inclusion and exclusion criteria, timing of diagnosis, and statistical methods—including adjustment for multiple comparisons and the absence of matching.
Thank you for recognizing this omission. We have clarified this point in lines 128–134 in the Materials and Methods section:
Inclusion criteria comprised all BCQP patients with available demographic, injury, and outcome data between 2000 and 2022. Patients with incomplete data were handled using pairwise deletion, consistent with BCQP registry practice. Diagnoses were identified by ICD-9/10 codes (Tables A1–A2). No matching was performed, as the analysis aimed to evaluate unadjusted population level associations across the entire cohort. Adjustment for multiple comparisons was not applied because each outcome represented a separate clinical endpoint. Table 1 compares four groups.
The title suggests that you investigated the presence of muscle wasting diagnosis pre-burn on outcomes. Please clarify whether the patients were diagnosed in hospital or were admitted with this diagnosis. I assume that this information might not be available from the database. Therefore, I would suggest adjusting the title.
Comment 4: Figures 1-3 are misleading because they show a timeline from 2013 to 2022 without indicating the incidences for the reported outcomes between 2013 and 2017 (incidence = 0).
Thank you for your feedback. We have clarified this point by adding the following explanation in the Results section (lines 146–150): Apparent absence of cases from 2013 to 2016 likely reflects limited adoption of muscle-wasting ICD codes within the BCQP during the early ICD-10 transition period, rather than a true lack of such patients. Reporting frequency for cachexia and protein-malnutrition codes increased steadily after 2016, consistent with broader national reporting of these diagnostic codes.
The fourth lines in Figures 1b, 2b, 3b are misleading. See comment 1.
Comment 5: Furthermore, the results section does not cover all relevant information presented in the tables, such as multiple sub-regression analyses that are not further discussed.
Thank you for this feedback. The BCQP analysis provided us a wealth of information regarding the effects of muscle wasting on burn outcomes, and we did our best to highlight the results we found most clinically significant while providing all results to the reader. We provided an explanation for our rationale in the Materials and Methods section in lines 128–134:
To enhance transparency and allow reproducibility of our methodology, extended variable definitions, coding criteria, and supplementary analyses are provided in the Appendix. Only material essential for interpretation of the primary and secondary outcomes has been retained within the manuscript, while additional non-essential tables are available online for readers who wish to examine the dataset in greater depth.
Comment 6: The discussion is convoluted and does not coherently build on the reported results.
Thank you for your feedback on the Discussion section of the manuscript. We added several sections to the Discussion section to improve readability, flow and understanding of the manuscript and study significance. Please refer to lines 274–277, 352-363 and 388–398 of the revised manuscript.
Author Response
Round 2, Comment 1a: Figure 1 suggests that there was no reporting of any muscle wasting conditions prior to 2017. Can you explain that?
Response: Thank you for your feedback. In the Materials and Methods section, we noted that “Some patients with undiagnosed or miscoded muscle-wasting conditions before 2017 were likely categorized in the no muscle wasting group, reflecting under-recognition of these diagnoses during earlier registry years” (lines 98-100). This under-recognition was likely due to ICD-10-CM codes being nationally implemented in 2015 and the specific code for sarcopenia (M62.84) being introduced in 2016. Overall, we observed that documentation of sarcopenia and cachexia (R64) appears in later BCQP records.
Round 2, Comment 1b: In the methods section you mention that: “Some patients with undiagnosed or miscoded muscle-wasting conditions before 2017 were likely categorized in the no muscle wasting group, reflecting under-recognition of these diagnoses during earlier registry years. (lines 95-97)” However, in the discussion you state that: “Moreover, there was an overall decrease in the number of patients with muscle wasting conditions as a total of the BCQP” (lines 318-319). The sentences contradict each other.
Response: Thank you for your observation. Please note that our observation of “an overall decrease in the number of patients with muscle wasting conditions as a total of the BCQP” (lines 298-299) in the current study versus the earlier study reflects a lower total percentage of BCQP patients with reported muscle-wasting conditions in the study timeframe. In the Materials and Method section, we clarify that (for the current study) “[b]ecause earlier years showed minimal or inconsistent coding for these diagnoses, we limited muscle-wasting analyses to 2017–2022 while retaining the full 2000–2022 BCQP cohort for context” (lines 95-98). Our initial study analyzed the BCQP patient population from 2000 up to 2017.
Round 2, Comment 1c: A suggestion would be to first report the prevalences of all muscle wasting conditions for each year, since the coding changed throughout your study period. Then, you could report the prevalences for the subcategories, beginning with e.g. the year when the distinct code was implemented. I would also suggest stating the population (no., years) that was used for these subcategories.
Response: Thank you for your suggestion. The prevalences of all muscle wasting conditions, stating the population and years, are listed in Figure 1 (line 164) and the prevalences for the muscle-wasting subcategories and all years of BCQP reporting are found in Figures 2 (line 174) and 3 (line 181).
Round 2, Comment 2: I would suggest strengthening your future outlook with more actionable recommendations for clinical practice.
Response: We appreciate your feedback and will consider incorporating more actionable recommendations for clinical practice in future studies.
Round 2, Comment 3: Table 1 compares four groups. The title suggests that you investigated the presence of muscle wasting diagnosis pre-burn on outcomes. Please clarify whether the patients were diagnosed in hospital or were admitted with this diagnosis. I assume that this information might not be available from the database. Therefore, I would suggest adjusting the title.
Response: Thank you for highlighting this ambiguity. In the discussion, we note that “the ICD codes that the patients are diagnosed with do not have an associated timing of diagnosis, making it difficult for the user to distinguish whether or not a given muscle wasting condition is preexisting or acquired during the acute stay of his or her burn care. For some terms, such as “sarcopenia” or “cachexia,” there is an association of chronicity which implies that the conditions are preexisting; however, with “protein malnutrition,” it is unclear if this condition is due to effects from burn hypermetabolism or preexisting factors unrelated to the burn injury” (lines 350-357).
We have since added the statement “In the BCQP, ICD codes do not have an associated timing of diagnosis, making it difficult for the user to distinguish whether or not a given muscle wasting condition is preexisting or acquired during the acute stay of his or her burn care” (lines 89-91) in the Materials and Methods section to clarify this point earlier in the manuscript. We also added the statement “ICD codes do not have an associated timing of diagnosis” (line 204) in the Table 1 title.
Round 2, Comment 4: The fourth lines in Figures 1b, 2b, 3b are misleading. See comment 1.
Response: Thank you for your observation. These tables provide more precise values for the data illustrated in Figures 1, 2 and 3. The percentages are outlined per year based on the patient populations of specific years and as a percentage of the total patient population since 2000. We have since corrected a typo in Figures 1, 2 and 3, noting that the solid line reflects the percentage of patients with muscle wasting/cachexia/protein malnutrition compared to the BCQP patient population since 2000 (rather than 2022).
Reviewer 4 Report
Comments and Suggestions for Authors
The authors have significantly improved the presentation and removed unnecessary sections. Thank you for addressing the suggestions — the manuscript is now much clearer and more coherent. I consider it ready for publication.
Author Response
Round 2, Comment 1: The authors have significantly improved the presentation and removed unnecessary sections. Thank you for addressing the suggestions — the manuscript is now much clearer and more coherent. I consider it ready for publication.
Response: Thank you for your comments and suggestions. Your feedback has greatly improved the readability and quality of our manuscript. Please note that we have added vertical lines to our tables to further enhance readability.